# Regucalcin Is a Potential Regulator in Human Cancer: Aiming to Expand into Cancer Therapy

**DOI:** 10.3390/cancers15225489

**Published:** 2023-11-20

**Authors:** Masayoshi Yamaguchi

**Affiliations:** Cancer Biology Program, University of Hawaii Cancer Center, University of Hawaii at Manoa, 701 Ilalo Street, Hawaii, HI 96813, USA; yamamasa11555@yahoo.co.jp

**Keywords:** regucalcin, cancer suppressor, cell signaling, cell proliferation, carcinogenesis, gene therapy

## Abstract

**Simple Summary:**

Regucalcin plays a multifunctional role in the regulation of cell function and expresses a repressive effect in the growth of normal cells. Of note, there is increasing evidence that regucalcin plays a potential role as a suppressor in several types of human cancer. Regucalcin expression is downregulated in the tumor tissues of cancer patients. Patients with higher levels of regucalcin in tumor tissues have shown longer survival. Overexpressed regucalcin suppresses the development of carcinogenesis. Extracellular regucalcin has been shown to suppress the proliferation of cancer cells. Delivery of the regucalcin gene may offer novel therapeutic benefits. This review discusses the potential role of regucalcin in the suppression of human cancer.

**Abstract:**

Regucalcin, a calcium-binding protein lacking the EF-hand motif, was initially discovered in 1978. Its name is indicative of its function in calcium signaling regulation. The *rgn* gene encodes for regucalcin and is situated on the X chromosome in both humans and vertebrates. Regucalcin regulates pivotal enzymes involved in signal transduction and has an inhibitory function, which includes protein kinases, protein phosphatases, cysteinyl protease, nitric oxide dynthetase, aminoacyl-transfer ribonucleic acid (tRNA) synthetase, and protein synthesis. This cytoplasmic protein is transported to the nucleus where it regulates deoxyribonucleic acid and RNA synthesis as well as gene expression. Overexpression of regucalcin inhibits proliferation in both normal and cancer cells in vitro, independent of apoptosis. During liver regeneration in vivo, endogenous regucalcin suppresses cell growth when overexpressed. Regucalcin mRNA and protein expressions are significantly downregulated in tumor tissues of patients with various types of cancers. Patients exhibiting upregulated regucalcin in tumor tissue have shown prolonged survival. The decrease of regucalcin expression is linked to the advancement of cancer. Overexpression of regucalcin carries the potential for preventing and treating carcinogenesis. Additionally, extracellular regucalcin has displayed control over various types of human cancer cells. Regucalcin may hold a prominent role as a regulatory factor in cancer development. Supplying the regucalcin gene could prove to be a valuable asset in cancer treatment. The therapeutic value of regucalcin suggests its potential significance in treating cancer patients. This review delves into the most recent research on the regulatory role of regucalcin in human cancer development, providing a novel approach for treatment.

## 1. Introduction

Cellular calcium plays a pivotal role as a second messenger in the hormone signal transduction mechanism of cell communication [1]. Its regulatory effect is potentially modulated by calcium-binding proteins, including calmodulin [2] and protein kinase C [3], which play critical roles as intracellular signaling molecules in the regulation of various cell functions [4]. In recent years, there has been increased attention on the functional pleiotropy of regucalcin, a novel calcium-binding protein, in cell signaling and disease [5,6]. Regucalcin was first discovered in 1978 as a calcium-binding protein lacking the EF-hand motif commonly found in calcium-binding domains [7,8]. This protein inhibits various calcium-dependant enzymes, such as Ca^2+^/calmodulin-dependent protein kinase and protein kinase C. Consequently, it serves a crucial role in cellular regulation as an inhibitory protein in the calcium signaling pathway [5,8]. Regucalcin’s nucleotide sequence was identified in 1993, and its gene symbol (*rgn*) is registered on PubMed [9]. The gene for regucalcin is located exclusively on the X chromosome, comprising of seven exons and six introns [5,10,11]. The regucalcin gene has been identified in over 15 vertebrate species, including humans, with the creation of the regucalcin family [12]. Later on, a protein identical to regucalcin, named senescence marker protein-30, was discovered [13,14].

The expression of the regucalcin gene is influenced by various physiological conditions, including different hormones and their associated signaling molecules. The promoter region of the gene has been thoroughly examined [11]. The regucalcin gene’s expression is enhanced by various transcription factors, such as calcium, AP-1, NF1-A1, RGPR-p117, β-catenin, HIF-1α, NF-κB, STAT3, SMADs, and the aryl hydrocarbon receptor (AHR) [15,16,17,18,19]. These factors have been identified as enhancers. Repressor elements have been found in the promoter region of the regucalcin gene. According to a study [20], SP1 acts as a suppressor in regucalcin gene expression. The discovery of RGPR-p117 was part of the transcription mechanism study of this gene [19], which is located on human chromosome 1q25.2 and is composed of 26 exons. Binding to a nuclear factor I (NFI) consensus motif TTGGC(N)6CC activates the regucalcin protein [19]. These transcription factors may contribute to cell regulation via regucalcin expression, as depicted in Figure 1.

Regucalcin mRNA and protein are expressed in various cell types and tissues [21]. Regucalcin functions as a regulator of signaling pathways in a variety of cells [5,6,8]. Regucalcin is critical to maintaining intracellular Ca^2+^ homeostasis by activating Ca^2+^ pump enzymes in the plasma membranes, mitochondria, and endoplasmic reticulum of cells, resulting in the reduction of cytoplasmic Ca^2+^ levels [5,8]. The cytoplasmic regucalcin is translocated to the nucleus of cells. Both cytoplasmic and nuclear regucalcins inhibit the activity of several Ca^2+^-dependent protein kinases, protein phosphatases, nitric oxide synthase, thiol protease, aminoacyl t-RNA synthetase, and protein synthesis involved in cell signaling [5,22]. Nuclear regucalcin hinders the Ca^2+^-activated process of DNA fragmentation and DNA and RNA synthesis. It also regulates gene expression, hence indicating its potential as a transcription factor [22]. As such, regucalcin exhibits the ability to regulate several molecules involved in signaling pathways within the cytoplasm and nucleus of cells. Overexpressed regucalcin has been shown to suppress cell proliferation enhancement [23] and apoptotic cell death [24] mediated by various signaling stimuli in cloned normal rat kidney proximal tubular epithelial cells and cloned rat hepatoma cells in vitro. Regucalcin is proposed to play a critical role as a suppressor in different signal transductions to maintain cell homeostasis for various stimuli [5]. Regucalcin may be involved in maintaining a physiological state and preventing metabolic disorders and related diseases, including osteoporosis [25], hyperlipidemia [25,26], diabetes [26], and renal failure [27].

The role of regucalcin in humans has yet to be fully understood, although the regucalcin gene and its protein sequences and crystal structure have been identified in the human species [12,28]. In recent years, evidence has increasingly suggested that regucalcin may play a role in suppressing human cancer [29,30,31,32,33]. Overexpressed regucalcin plays a critical role in suppressing cell growth by controlling various signaling processes associated with cell proliferation, repressing oncogene expression, and stimulating tumor suppressor gene expression in various types of human cancer cells [34,35,36,37,38,39,40]. Furthermore, regucalcin gene expression and protein levels are downregulated in human cancer tissues as per multiple gene expression profiling and proteomics analyses [34,35,36,37,38,39,40]. Higher expression of regucalcin in tumor tissues has been demonstrated to extend the survival of cancer patients [34,35,36,37,38,39,40]. It has been suggested that regucalcin may have a potential role in suppressing human cancers. This review presents recent advancements in understanding the role of regucalcin as a tumor suppressor in human cancer.

## 2. Overexpressed Regucalcin Suppresses Cell Growth In Vitro

Regucalcin has been shown to suppress the proliferation of various cell types, including normal and cancer cells. Overexpressed regucalcin suppressed proliferating cells via the mechanism involving the inhibition of the cell signaling process, independent of apoptotic cell death in vitro [23]. The overexpression of regucalcin blocked the pathway of the G1 progression and G2/M phase cell cycle due to the inhibition of cyclin-dependent kinases (cdc2, cdk2m, and cdk5) in proliferating cells [23]. Furthermore, overexpressed regucalcin increased the expression of p21, an inhibitor of cyclin-dependent kinases (cdk) [23]. Overexpressed regucalcin did not aeffect the expression of cdc2a and chk2 (checkpoint kinase 2) mRNAs [23]. Thus, regucalcin has demonstrated the ability to hinder the growth of diverse cell types, including those implicated in cancer.

The expressions of regucalcin are decreased in liver cancer cells compared to normal livers [23,32]. Overexpression of regucalcin suppressed the growth of H4-II-E cells and HepG2 cells in vitro [23,32]. The underlying mechanism by which the overexpressed regucalcin suppresses cell growth has been implicated in the inhibition of mitogen-activated protein (MAP) kinase, Ca^2+^/calmodulin-dependent protein kinases, protein kinase C, and various protein phosphatases, including protein tyrosine phosphatase and protein serine/threonine phosphatases in the cytoplasm and nuclei of cells in vitro [23,32]. In addition, regucalcin suppresses cytoplasmic protein synthesis by inhibiting aminoacyl t-RNA synthase activity in the cytoplasm of cells [23,32], resulting in protein loss in cancer cells. Of note, overexpressed regucalcin suppressed the enhancement of nuclear DNA synthesis in rat hepatoma H4-II-E cells [23,32]. Thus, regucalcin has been shown to suppress proliferation in normal and cancer cells, suggesting a physiological and pathophysiological role in the control of cell overproliferation.

As an underlying mechanism, overexpressed regucalcin has been demonstrated to suppress tumor-related gene expression, such as c-myc, c-fos, c-jun, c-src or Ha-ras, which are tumor stimulator genes [23,32]. It is worth noting that c-src is an oncogene [41]. It is noteworthy that overexpression of regucalcin inhibits the mRNA levels of p53 and Rb, which are tumor suppressor genes [42]. Additionally, p53 stimulates the gene expression of p21, which is an inhibitor of protein kinases that are related to the cell cycle. These genes have been demonstrated to increase cancer cell growth in vitro [42]. This, in turn, induces cell cycle arrest [23,32], providing evidence that the alteration of these molecules may play a role in controlling cell proliferation in vitro. Downregulation of the regucalcin gene expression in cancer cells could lead to tumorigenesis with rapidly proliferating cells [23,32].

As stated previously, Figure 2 provides a summary of the molecules and genes that regucalcin targets to inhibit the growth of both normal and cancer cells. Regucalcin plays a critical role in the suppression of cell proliferation by regulating multiple signaling processes.

## 3. The Repressive Role of Regucalcin in Carcinogenesis In Vivo Models

Regucalcin is involved in the control of cell growth and carcinogenesis in vivo models. The hepatocytes of rat liver are normally *quiescent* in vivo. Partial hepatectomy liver tissues increase the proliferation of hepatocytes to restore the removed liver tissue [43]. The regenerating liver is a good model for the cell proliferation of liver tissue in vivo [43]. Regucalcin may help to control the growth of the regenerating liver and the development of hepatocarcinogenesis in animal models in vivo [44]. Regucalcin mRNA expression was increased in regenerating rat liver after partial hepatectomy [44], suggesting an involvement in suppressing liver cell overproliferation with the regenerating liver. Enhanced endogenous regucalcin increased the activities of the plasma membrane (Ca^2+^-Mg^2+^)-ATPase and nuclear Ca^2+^-ATPase to maintain cytoplasmic and nuclear calcium levels [44]. Mechanistically, regucalcin suppressed the activities of Ca^2+^/calmodulin-dependent protein kinases, protein kinase C, protein phosphatases, and the synthesis of protein, DNA, and RNA in the nucleus, which is enhanced with proliferating cells in the regenerating liver, to control liver cell overproliferation in vivo [44]. Thus, endogenous regucalcin has been shown to play a role in controlling liver regeneration after partial hepatectomy in vivo.

Liver carcinogenesis is induced in vivo in rats by continuous feeding of the basal diet containing 0.06% 3′-methyl-4-dimethylaminoazobenzene [45]. In this animal model, the study identified a decrease in regucalcin mRNA expression in rat hepatoma tissues compared to non-tumor liver tissues. Additionally, a specific increase in c-myc mRNA was observed in the tumor tissues [45]. Notably, the mutation of the regucalcin gene was not found in the tumor tissues [45]. This finding was the first to implicate regucalcin in cancer in vivo. In other animal models, new markers of liver carcinogenesis were identified in the pre-neoplastic foci of the liver of rats partially hepatectomized in vivo with both diethylnitrosamine and 2-acetylaminofluorene, which induces liver carcinogenesis [46]. Biomarkers, including transaldolase, aflatoxin B1 aldehyde reductase, and gamma-glutamylcysteine synthetase, were identified as upregulated genes in hepatocellular carcinoma (HCC) [46]. Specifically, regucalcin expression was shown to be downregulated in HCC, especially in the early stages of carcinogenesis [46]. Additionally, in vivo studies with CuZn superoxide dismutase (CuZnSOD, Sod1)-deficient mice have demonstrated their ability to induce HCC [47]. In this animal model, a considerable decrease in regucalcin was found in −/− samples with HCC development [47]. This indicates that the downregulation of regucalcin gene expression is linked with carcinogenesis in vivo animal models, supporting the notion that regucalcin could potentially suppress tumorigenesis in rats.

Regucalcin is expressed not only in the liver but also in the prostate and mammary glands of both rats [48] and humans [49]. Regucalcin gene expression in these tissues was downregulated by sex steroid hormones, including 17β-estradiol [50,51]. Interestingly, overexpression of regucalcin led to the suppression of cell proliferation in the prostate of regucalcin transgenic rats in vivo [50]. In addition, it has been shown that regucalcin gene expression in the prostate decreases with aging [50]. This decrease was not seen in transgenic rats overexpressing regucalcin [50]. Increasing age leads to decreased glutathione activity and antioxidant capacity in the prostate. These decreases were found to be prevented in regucalcin transgenic rats [50]. Thus, overexpression of regucalcin may have a protective effect against age-related pathologies, such as prostate cancer [50].

In addition, regucalcin is involved in mammary malignancy using regucalcin transgenics in vivo [51]. 7,12-dimethylbenz[α]anthracene, a carcinogen [51], was administered to wild-type and regucalcin transgenic rats [51]. Regucalcin expression decreased with the histologic grade of breast infiltrating ductal carcinoma (IDC) [51]. This study implies that breast cancer progression is associated with a decline in regucalcin [51]. Furthermore, regucalcin transgenic rats demonstrated a lower incidence of carcinogen-induced mammary tumors through a reduction in cell cycle inhibitors and an increase in apoptosis inducers [51]. Thus, elevated levels of regucalcin were demonstrated to obstruct tumor growth in mammary glands induced by carcinogens in vivo [51].

As stated previously, regucalcin has a protective role in the overproliferation of regenerating rat liver after partial hepatectomy in vivo models. Additionally, the expression of regucalcin decreases in animals with cancer and contributes to the development of liver, prostate, and mammary gland carcinogenesis in in vivo animal models.

## 4. The Role of Regucalcin in the Suppression of Human Cancer

The gene for regucalcin is expressed in the human species [12]. Characteristics of the human regucalcin gene have been discovered. Transcript heterogeneity is found in the human regucalcin gene [52], although the significance of transcript heterogeneity of the human gene for regucalcin is unknown. Alternatively spliced variants of the regucalcin gene have been found in various normal and tumor tissues of human subjects, although they have not been observed in animals, including rats and mice [53]. The different variants of regucalcin mRNA, both full-length and alternatively spliced, were expressed in various human normal tissues. However, they were suppressed in tumor tissues such as hepato-cellular carcinoma, renal transitional cell carcinoma, brain malignant meningioma, and lung non-small cell carcinoma [53]. Other tumor tissues may also contain these variants. Overexpressed regucalcin has demonstrated its potential in suppressing the growth of human cancer cells [34,35,36,37,38,39,40]. However, overexpression of the spliced variant proteins did not have a significant suppressive effect on the growth of cancer cells [53]. Further studies are required to determine the significance of the spliced variant proteins in the regulation of human cells.

Regucalcin may potentially suppress human cancer [23,32]. Overexpressed regucalcin plays a suppressive role in human cancer cell growth by suppressing various signaling processes associated with cell proliferation by repressing oncogene expression and stimulating tumor suppressor gene expression in various types of human cancer cells [34,35,36,37,38,39,40]. In addition, the downregulation of the regucalcin gene and its protein expressions in human cancer tissues has been demonstrated by multiple gene expression profiling and proteomics analyses [34,35,36,37,38,39,40]. Notably, patients with various types of human cancer experience prolonged survival with an increased expression of regucalcin in tumor tissues [34,35,36,37,38,39,40], as demonstrated in Figure 3.

### 4.1. Liver Cancer

Regucalcin is highly expressed in both animal and human liver and is suppressed in liver cancer of humans. Hepatocellular carcinoma (HCC) is the most common primary liver cancer, which accounts for most of the prevalent malignancies and the primary cause of cancer-linked deaths [54,55]. HCC usually arises in cirrhosis conditions, a chronic and diffuse liver ailment that occurs due to continuous liver regeneration and injury [56,57]. HCC cases are also associated with chronic viral infections, such as hepatitis B or hepatitis C [58,59,60,61,62]. External stimuli initiate a multistep process leading to hepatocellular carcinogenesis.

Regucalcin plays a critical role as a suppressor in human liver cancer, as depicted in Figure 3 [34]. When comparing liver expression levels of regucalcin in 35 healthy individuals and 47 patients with HCC, a decrease in its expression was observed among HCC patients using the Gene Expression Omnibus (GEO) database (GSE45436) and the Human Protein Atlas, as reported by [34]. Moreover, a clinical evaluation was conducted between 81 HCC patients with higher regucalcin expression and 81 HCC patients with lower regucalcin expression [34]. The results showed that reduction of regucalcin expression was associated with poor prognosis in HCC patients [34]. Higher regucalcin gene expression was associated with prolonged survival in HCC patients [34]. It can be hypothesized that downregulated regucalcin expression plays a role in the development of carcinogenesis in human HCC cells. Regucalcin has potential as a suppressor of human HCC. Translational studies have demonstrated that the overexpression of regucalcin inhibits the proliferation, cell death, and migration of human liver cancer HepG2 cells in vitro [34]. The overexpression of regucalcin suppressed the G1 and G2/M phase cell cycle and proliferation of HepG2 cells by inhibiting several signaling pathways comprising Ras, Akt, MAP kinase, SAPK/JNK, NF-κB p65, and the expression of the oncogenes c-fos and c-myc while increasing the levels of tumor suppressors p21, p53, and Rb [34]. Remarkably, the overexpression of regucalcin reduced the levels of β-catenin, which is a major oncogenic molecule in HCC [34]. Mutation of the β-catenin gene is detected in patients with HCC and has been found to alter the expression of β-catenin target genes such as glutamate synthetase, axin2, lect2, and regucalcin [63]. Furthermore, in vitro studies indicate that the growth of HepG2 cells is promoted by the calcium channel agonist Bay K 8644 [64]. However, this promotion is hindered by regucalcin overexpression in HepG2 cells [64]. Thus, the elevated expression of regucalcin demonstrated inhibition of calcium signaling related to the growth of liver cancer cells.

Furthermore, a tissue microarray confirmed that regucalcin is preferentially expressed in the normal human liver [65]. The level of regucalcin was reduced in HCC tissues compared to non-tumor tissues [65]. The downregulation of regucalcin in HCC was suggested to be mediated by DNA methylation [65]. This study provides additional support for the notion that regucalcin serves as a possible clinical prognostic marker and therapeutic target for HCC [65].

Serum regucalcin may be clinically significant as a biomarker in human liver cancer [66,67,68,69]. Regucalcin may be an HCC-associated antigen [66]. A total of 175 instances of HCC serum were examined to ascertain the presence of anti-regucalcin antibodies [66]. Specifically, 22 cases were confirmed as positive [66]. In addition, the regucalcin and anti-regucalcin antibody levels in the serum of 143 patients with HCC were compared to those in serum samples from 137 patients with chronic hepatitis, 51 individuals with liver cirrhosis, and 165 healthy control participants [67]. The rate of positivity for anti-regucalcin antibodies in HCC patients was higher than that in the chronic hepatitis group and the liver cirrhosis group [67]. This implies that the levels of anti-regucalcin antibodies in the serum could serve as a unique biomarker for diagnosing HCC [67]. It is worth noting that significant outcomes have been recorded in patients with alpha-fetoprotein (AFP) negativity [67].

Liver injury and hepatitis can lead to the development of hepatocellular carcinoma (HCC) in humans. Regucalcin in serum is a potential biomarker for detecting hepatitis. A study found that regucalcin was released in the serum of human subjects with hepatitis [69]. The study collected serum samples from 42 individuals diagnosed with liver disease. The serum regucalcin concentration in all patients was significantly higher than that in the serum of normal subjects (10 persons) without hepatitis [69]. Notably, the serum levels of glutamine-oxaloacetate transaminase (GOT) and glutamate-pyruvate transaminase (GPT) in 18 patients with liver damage were within normal range [69]. The assessment of serum regucalcin served as an effective diagnostic tool for chronic liver injury, presenting a lower level of serum GOT and GPT activities. Serum regucalcin exhibits potential as a biochemical marker for hepatitis due to its potential sensitivity.

Furthermore, serum regucalcin was determined in 47 chronic hepatitis B patients, 91 hepatitis B virus (HBV)-related acute-on-chronic liver failure (HBV-ACLF) patients, and 33 healthy controls [70]. The median serum regucalcin concentrations in the HBV-ACLF and chronic hepatitis B patients were much higher than in the healthy controls. Serum regucalcin may be more useful as a marker of liver injury [69,70].

Interestingly, the development of a regucalcin antigen vaccine for HCC immunotherapy has been investigated. Dendritic cells (DC) pulsed with recombinant regucalcin were shown to induce cytotoxic T lymphocytes (CTLs) against liver cancer cells in vitro [71]. Regucalcin and the heat shock protein GP96 were subcloned into lentiviruses and transfected into DCs from healthy donors [71]. Regucalcin plus GP96 effectively stimulated the proliferation of T cells compared to the control treatment [71]. In the constructed liver cancer model, the regucalcin plus GP96 group showed a better effect [72]. In addition, DCs transduced with LV-regucalcin were shown to enhance specific T-cell immune responses against murine hepatocarcinoma cells in vitro and in vivo [71,73]. These studies may lead to the development of a DC-based regucalcin antigen vaccine for HCC immunotherapy [71,73].

As mentioned above, it has been demonstrated that the downregulation of liver regucalcin may lead to hepatocellular carcinogenesis (HCC) and that this protein plays a critical role as a suppressor of HCC. In addition, serum regucalcin may play a clinical role as a biomarker for the diagnosis of HCC and as an antigen for immunotherapy. 

### 4.2. Lung Cancer

Human lung cancer is categorized into two types: small-cell lung cancer (SCLC) and non-small cell lung cancer (NSCLC). NSCLC accounts for over 80% of all cases and is a significant factor in malignancy-related deaths and long-term survival rates, according to [74,75,76,77,78,79,80,81]. Additional therapeutic measures may be required for lung cancer sufferers. Notably, the decreased expression of regucalcin contributes to the growth of human lung cancer [37]. Gene expression and survival data from 204 patients with lung adenocarcinoma were retrieved from the GEO database (GSE31210) to carry out outcome analysis [37]. The results revealed that regucalcin expression was downregulated in lung cancer patients. Moreover, higher regucalcin gene expression led to prolonged survival in lung cancer patients. In translational studies, it was observed that the overexpression of regucalcin suppressed proliferation, cell death, and migration in human lung adenocarcinoma NSCLC A549 cells in vitro [37]. In the underlying mechanism, regucalcin overexpression led to cell cycle arrest in A549 cells by suppressing multiple signaling pathways such as Ras, Akt, MAP kinase, and SAPK/JNK [37]. Additionally, the overexpressed regucalcin reduced the oncogenes c-fos and c-myc while upregulating the tumor suppressors p53 and Rb [37].

Note that regucalcin and survivin are involved in the process of aging and overcoming aging and are epigenetically modified in NSCLC tissues, as demonstrated by analysis of a methylome bead chip and corresponding transcriptome [82]. The study has shown that a higher expression of survivin is associated with lower survival rates in adenocarcinoma patients, while a higher expression of regucalcin is correlated positively [82]. Epigenetic reprogramming in NSCLC may increase survivin expression and decrease regucalcin expression [82]. Survivin and regucalcin could be potential predictive markers in NSCLC [82].

The study investigated the role of regucalcin in NSCLC, using tumor and normal tissue from 341 NSCLC patients [83]. Results indicated that regucalcin expression was significantly reduced in NSCLC tissue compared to non-tumor tissue [83]. Additionally, Kaplan-Meier survival analysis demonstrated that patients with lower regucalcin expression had poorer overall survival compared to patients with higher regucalcin expression [83]. Translational studies indicate that overexpressed regucalcin inhibits the proliferation of A549 and H1299 cells in vitro and in tumor xenografts. Additionally, it downregulates the expression of c-myc, cyclinD1 protein, and histone deacetylase 4 (HDAC4) [83]. Notably, HDAC4 overexpression reverses the inhibition of NSCLC mediated by regucalcin both in vitro and in vivo [83]. Regucalcin was shown to inhibit NSCLC proliferation by reducing HDAC4 expression, suggesting a possible mechanism by which regucalcin suppresses NSCLC.

Interestingly, recent research indicates that regucalcin may play a significant role in the tumor immunological microenvironment of lung squamous cell carcinoma (LUSC) [83]. By utilizing ESTIMATE and CIBERSORT algorithms and the Tumor Immune Estimation Resource database, the correlation between regucalcin and immune cells was established [83,84]. Furthermore, regucalcin expression was found to be reduced in the tumor tissues of LUSC patients [84]. It is notable that the expression of regucalcin was greatly involved in immunobiological processes [84]. The results of this study demonstrate a positive correlation between regucalcin expression and specific immune-infiltrating cells in the lung tumor microenvironment. Regucalcin’s potential role in this context suggests that it may serve as a valuable biomarker for assessing immunotherapy efficacy and patient prognosis.

### 4.3. Prostate Cancer

Prostate cancer, a malignancy which commonly spreads to bone, ranks among men’s top malignant growths, and is the second leading cause of cancer-related death for men [85,86,87,88]. This condition particularly infiltrates bones, leading to severe complications like excruciating pain, fractures, spinal cord compression, and bone marrow suppression [89,90,91,92,93,94,95]. The most prevalent cause of death among patients afflicted with prostate cancer is the metastatic spread of the tumor or disease recurrence. Regucalcin plays a role in the suppression of prostate cancer (Figure 3) [40]. This study elucidates the role of regucalcin in prostate cancer tumor malignancy [40]. The expression of regucalcin was lower in metastatic tumors than in primary tumors [40]. Prostate cancer patients with higher regucalcin expression had a prolonged progression-free survival compared to those with lower regucalcin gene expression [40]. Additionally, the translational study demonstrated that in vitro overexpression of regucalcin significantly decreased colony formation and cell growth in bone metastatic human prostate cancer PC-3 and DU-145 cells [40]. Overexpression of regucalcin increased the levels of p53, Rb, and p21, while decreasing the levels of Ras, PI3 kinase, Akt, MAP kinase, and transcription factors, including NF-κB p65, β-catenin, and signal transducer and activator of transcription 3 (STAT3). This leads to the control of cell growth and suggests that regucalcin has a suppressive effect on lung cancer by regulating the expression of various molecules in different signaling pathways [40].

Regucalcin promotes dormancy of prostate cancer [96]. The study confirmed that higher levels of regucalcin were associated with longer recurrence-free and overall survival of prostate cancer patients. An ectopic expression system induced dormancy in vivo of prostate tumor cells using doxycycline-inducible regucalcin expression [96]. Interestingly, the study found that knocking down regucalcin in LNCap cells, a human prostate cancer cell line, led to their increased growth in the tibia of mice [96]. Regucalcin was found to promote tumor dormancy through various mechanisms, including activation of p38 MAP kinase, decrease in Erk signaling, and inhibition of FOXM1 expression [96]. Furthermore, regucalcin was shown to suppress angiogenesis by raising secretory miR-23c levels in exosomes [96]. Therefore, this research supports the crucial role of regucalcin in maintaining the quiescence of the prostate [96].

Notably, an overexpression of regucalcin was found to suppress the migration and invasion of bone metastatic human prostate cancer cells in vitro [97]. Mechanistically, overexpression of regucalcin led to decreased levels of key metastasis proteins, including Ras, Akt, MAP kinase, RSK-2, mTOR, caveolin-1, and integrin β1 [97]. Additionally, the invasion of prostate cancer cells was promoted by co-culturing with preosteoblastic MC3T3-E1 or preosteoclastic RAW264.7 cells [97]. Thus, the overexpression of regucalcin in prostate cancer cells prevented aberrant bone cell differentiation. It is noteworthy that regucalcin overexpression also inhibited TNF-α production in prostate cancer cells [97].

As previously stated, regucalcin suppresses human prostate carcinogenesis. When there is a decrease in regucalcin in human prostate cells, it leads to increased prostate cancer metastasis. Conversely, a higher expression of regucalcin in prostate cancer cells inhibits their migration, invasion, and bone metastatic activity.

### 4.4. Breast Cancer

Breast cancer is the most prevalent malignant form and the primary reason for death due to cancer among women in the United States. The malignancy is prone to metastasize to the bone [72,98,99,100,101,102], inducing agonizing pathological fractures, pain, and hypercalcemia [72,98,99,100,101,102]. The invasion of the tumor in the bone tissue corresponds to the activation of osteoclasts and osteoblasts, which are the main bone cells [100,102,103,104]. For patients of breast cancer with bone metastases, bisphosphonates or denosumab are the standard care treatment options [105].

The expression pattern of regucalcin was compared in human, canine, and feline mammary carcinomas [106]. Regucalcin was specifically observed in neoplastic mammary epithelial cells, while its expression was low in normal mammary gland tissues or well-differentiated adenoma tissues. This study provides valuable information to comprehend the expression of regucalcin in different stages of mammary carcinoma and indicates its usefulness as a pan-species diagnostic marker.

The study examined the role of regucalcin in human breast cancer patients by analyzing data from the GEO database (GSE6532). The goal was to compare clinical outcomes between 44 patients expressing higher levels of regucalcin and 43 patients with lower expression (see Figure 3) [36]. It was found that regucalcin expression is down-regulated in breast cancer patients [36], and patients with higher regucalcin levels had longer relapse-free survival [36]. Additionally, the translational study showed that overexpressing regucalcin led to cell cycle arrest and inhibited the proliferation of bone metastatic human breast MDA-MB-231 cells in vitro [36]. Mechanistically, the overexpression of regucalcin suppressed various signaling pathways, such as Akt, MAP kinase, SAPK/JNK, NF-κB p65, and β-catenin, while increasing the tumor suppressor p53 and decreasing K-ras, c-fos, and c-jun [36]. Additionally, research has demonstrated that the coculture of regucalcin-overexpressing MDA-MB-231 cells and mouse bone marrow cells inhibited enhanced osteoclastogenesis and suppressed mineralization in vitro [36]. Furthermore, higher expression of regucalcin was shown to suppress the growth and bone metastatic activity of breast cancer cells, potentially contributing to relapse-free survival in patients.

### 4.5. Pancreatic Cancer

The pancreas comprises two types of cells—endocrine and exocrine cells. Pancreatic ductal adeno-carcinoma (PDAC) accounts for about 90% of all pancreatic cancers [107,108,109]. As a highly aggressive malignancy [110,111,112], currently available therapies only provide limited treatment options for pancreatic cancer patients [110,111,112]. K-ras mutations are present in the majority (90%) of pancreatic cancers [113,114,115,116]. The function of regucalcin in human pancreatic cancer has been clarified through assessment of its expression levels in both normal pancreatic tissue and pancreatic ductal adenocarcinoma (PDA) among human subjects [35]. Microarray analysis revealed downregulation of regucalcin expression levels in pancreatic tissue from 36 PDA patients as compared to tissue from 36 normal pancreases, as determined through the utilization of the GEO database (GSE15471) [35]. The survival of pancreatic cancer patients with increased regucalcin gene expression was extended (Figure 3). The study found that overexpression of regucalcin inhibited proliferation, cell death, and migration in human pancreatic cancer MIA PaCa-2 (K-ras mutated) cells, which are resistant to drug and radiation therapy [35]. The suppressive effects of regucalcin on cell proliferation were not dependent on cell death. The overexpression of regucalcin inhibits various signaling pathways, including Akt, MAP kinase, SAPK/JNK, K-ras, c-fos, and c-jun [35]. Intriguingly, regucalcin overexpression boosts the levels of p53 protein, a tumor suppressor [35]. Therefore, regucalcin demonstrates potential as a tumor suppressor in human pancreatic cancer. The decrease in regucalcin expression may lead to the onset of carcinogenesis in pancreatic tissues.

### 4.6. Colorectal Cancer

Adenocarcinoma is the primary malignancy affecting the colon and rectum [117]. Colorectal cancer ranks third among the most frequently diagnosed cancers [118,119], with an average five-year survival rate of only 55% [119]. Despite novel therapeutic strategies being developed, the prognosis for CRC remains dismal [120,121,122,123]. Identification and characterization of innovative biomarkers could offer scope for prolonging the survival in colorectal cancer. Mutations in the KRAS gene have been found in over 40% of tumors displaying genetic and epigenetic alterations [124,125,126,127]. Regucalcin plays a role in the inhibition of human colorectal cancer [38]. To examine the outcomes, regucalcin gene expression and survival data of 62 patients from the GEO database (GSE12945) were analyzed (Figure 3) [38]. Regucalcin expression was reduced in colorectal cancer patients [38]. Prolonged survival among colorectal cancer patients is significantly linked with elevated regucalcin gene expression in their tumor tissue [38]. In vitro translational findings reveal how overexpressed regucalcin can lead to the suppression of colony formation and proliferation of human colorectal-cancer-derived RKO cells [38]. Mechanistically, this overexpression hinders the cell cycle of RKO cells by inhibiting crucial signaling pathways related to Ras, Akt, MAP kinase, and SAPK/JNK [38]. Notably, the overexpression of regucalcin increased the levels of tumor suppressors p53 and Rb, as well as the cell cycle inhibitor p21 [38]. Moreover, the overexpression of regucalcin repressed the transcription factors c-fos, c-jun, NF-κB p65, β-catenin, and STAT3 [38], indicating that regucalcin targets various signaling molecules. This study indicates that regucalcin plays a crucial role as a suppressor in human colorectal cancer. Higher levels of regucalcin with gene delivery could potentially serve as a novel therapy for colorectal cancer.

### 4.7. Kidney Cancer

Renal cell carcinoma (RCC) is a type of cancer that develops in the lining of kidney tubules [128,129]. RCC is the second leading cause of death among urological malignant neoplasms [130,131,132,133]. Clear cell RCC is the most frequent histological subtype, representing around 80–90% of all RCCs [129]. The treatment of RCC employs FDA-approved agents such as mammalian target of rapamycin (mTOR), vascular endothelial growth factor (VEGF), platelet-derived growth factor (PDGF), and their corresponding receptors VEGFR and PDGFR [134,135]. Nevertheless, the therapeutic benefits of these inhibitors could be curbed because of the emergence of drug-resistant phenotype [136,137,138,139,140]. To investigate the role of regucalcin in RCC, we obtained data from kidney cortex tissues of clear-cell RCC patients through the GEO database (GSE36895). Our analysis revealed downregulation of regucalcin expression in RCC tumor tissues. Interestingly, an association between higher regucalcin gene expression and prolonged survival in clear-cell RCC patients was observed (see Figure 3) [39]. The translational study showed that overexpression of regucalcin inhibited colony formation and proliferation of human clear-cell RCC A498 cells in vitro [39]. Mechanistically, regucalcin overexpression resulted in G1 and G2/M phase arrest of A498 cells by suppressing multiple signaling components such as Ras, PI3 kinase, Akt, and MAP kinase [39]. The enhanced expression of regucalcin significantly increased the levels of tumor suppressors, namely p53, Rb, and the cell cycle inhibitor, p21 [39]. Therefore, regucalcin suppresses the advancement of human RCC by aiming at various molecules involved in the intra.

### 4.8. Cervical Adenocarcinoma

Cervical cancer is a tumor that has a high morbidity and mortality rate [141,142,143,144]. Compared to squamous cell carcinoma, cervical cancer has a higher rate of ovarian metastasis [145,146,147]. Due to its safety and specificity, gene therapy is emerging as a potential therapeutic option for cervical cancer [148]. The transfection of lentivirus-mediated regucalcin into HeLa cells has been found to increase regucalcin expression and significantly reduce cell proliferation, invasion, and promote cell cycle arrest at the G2/M phase [149]. A higher expression of regucalcin results in a decreased level of β-Catenin, p-glycogen synthase kinase-3β (GSK-3β), and matrix metalloproteinases (MMPs) -3, -7, and -9 [150]. E-cadherin and GSK-3beta levels were increased through regucalcin overexpression [149]. Regucalcin exhibited inhibition of cervical cancer tumorigenesis through mechanisms involving Wnt/β-catenin signaling and epithelial-mesenchymal transition, which suppress tumor proliferation and metastasis [149]. Notably, lentivirus-mediated siRNA downregulation of regucalcin was found to promote cell proliferation, migration, and invasion. Thus, studies have demonstrated that regucalcin plays a suppressive role in the development of cervical cancer in humans. Upregulating the expression of regucalcin may provide a potential therapeutic approach for treating cervical cancer.

### 4.9. Melanoma

Melanoma is a highly aggressive type of skin cancer [151,152]. Breslow’s thickness (T stage) [153] is one of the key factors that determines prognosis and treatment for locally advanced melanoma. This measure is based on the thickness of the main tumor in millimeters. The discovery of new biomarkers could have clinical implications. Biomarkers such as serum lactate dehydrogenase (LDH) and S100B could be linked to clinical stage and tumor progression [154]. Regucalcin may be a significant biomarker in human melanoma. Affinity proteomic assays were employed to profile serum samples from patients with malignant melanoma to identify proteins present in the bloodstream linked with melanoma stage or recurrence. The analysis encompassed 149 serum samples from patients with malignant melanoma. Notably, patients with recurrent tumors and high Breslow’s were found to have lower serum levels of regucalcin and syntaxin 7 (STX7) than those with low thickness and no recurrence [154]. Regucalcin shows potential as a new biomarker for human melanoma.

### 4.10. Osteosarcoma

Osteosarcoma originates in the bone, and there have been few advances in survival and treatment of metastatic disease [155,156,157]. Chondrosarcoma is the most common bone sarcoma in adults [155,156,157]. Pain is the most common presenting symptom in patients with bone tumors [156]. The primary tumor of osteosarcoma is surgically resected [157,158,159,160]. The involvement of regucalcin in human osteosarcoma has been investigated using Saos-2 human osteosarcoma cells in vitro [161]. The overexpression of regucalcin was found to suppress the growth of Saos-2 cells in vitro [161]. The suppressive effects of overexpressed regucalcin on the proliferation of Saos-2 cells were suggested to involve the suppression of signaling pathways, including PI3K/Akt, extracellular signal-regulated kinase (ERK)/MAP kinase, and a protein kinase C, by using each specific inhibitor [162,163,164]. In addition, the suppressive effects of overexpressed regucalcin on the proliferation of Saos-2 cells may be involved in the regulation of nuclear functions from the results obtained by using an inhibitor of RNA polymerase II-dependent transcriptional activity [165] and gemcitabine, an antitumor agent that induces nuclear DNA damage [166]. In addition, overexpressed regucalcin decreased the levels of several proteins involved in signaling pathways related to Ras, PI3K, Akt, MAP kinase, STAT3, β-catenin, and NF-κB p65, and increased the levels of p53, Rb, and p21 in Saos-2 cells [161]. Thus, the suppressive effects of regucalcin on the proliferation of human osteosarcoma Saos-2 cells may be exerted through the regulation of various signaling pathways. Regucalcin may contribute as a suppressor in human osteosarcoma.

### 4.11. Ovarian Cancer

Ovarian cancer has the highest mortality rate among gynecologic malignancies and an average five-year survival rate for tumor patients [167,168,169,170,171,172,173,174,175,176]. It is a complex and heterogeneous malignant disease. Various risk factors, such as nulliparity, infertility, endometriosis, obesity, and advanced age are associated with the underlying mechanism leading to ovarian cancer. Primary surgery and drug therapy are commonly utilized treatment methods for ovarian cancer. The study has demonstrated the inhibitory effect of regucalcin on cell growth in human ovarian cancer SK-OV-3 cells, which have shown resistance to cytotoxic cancer drugs [177]. Overexpression of regucalcin repressed the colony formation and proliferation of SK-OV-3 cells through an independent mechanism of cell death [177]. Overexpression of regucalcin decreased the levels of Ras, Akt, MAP kinase, NF-κB p65, β-catenin, and STAT3, while elevating the levels of tumor suppressors p53 and Rb, as well as the cell cycle inhibitor p21 [177]. Notably, the proliferative effects of epidermal growth factor (EGF) on cell proliferation were inhibited by the overexpression of regucalcin in SK-OV-3 cells [177]. Thus, overexpressed regucalcin may repress cell proliferation by targeting diverse signaling pathways, including EGF signaling. This study suggests the involvement of regucalcin as a suppressor in ovarian cancer.

## 5. The Suppressive Role of Extracellular Regucalcin in the Cancer Microenvironment

Regucalcin is expressed in various organ cells, including the liver and kidney, in both humans and animals [21,69]. Physiological levels of regucalcin in human serum are 1 nM [21,69]. It is worth noting that isolated regucalcin has been found to bind to plasma membranes, and regulate the activity of the pump enzyme (Ca^2+^-Mg^2+^)-adenosine triphosphatase [178]. As a result, extracellular regucalcin has been suggested to play a role in regulating cell function. In recent years, mounting evidence suggests that extracellular regucalcin suppresses the growth of various types of cancer cells. Additionally, regucalcin may have therapeutic potential in the development of cancer within the microenvironment, presenting a novel strategy for cancer therapy. This section delves into the role of extracellular regucalcin as a cancer cell suppressor.

In modeled human liver cancer HepG2 cells, extracellular regucalcin was found to have suppressive effects on cell growth in vitro [179]. In this study, regucalcin was used at physiological levels (0.01–10 nM). Extracellular regucalcin did not affect apoptotic cell death [180]. In addition, extracellular regucalcin suppressed the colony formation of HepG2 cells in vitro [180]. This study demonstrated that extracellular regucalcin exerts an inhibitory effect on the growth of human liver cancer cells.

Extracellular regucalcin is also shown to suppress the growth of human pancreatic cancer MiaPaCa-2 cells in vitro [179]. The proliferation of MiaPaCa-2 cells was suppressed by culturing with the addition of regucalcin [179]. The suppressive effects of regucalcin on cell proliferation were not potentiated by the presence of various signaling inhibitors, including TNF-α, Bay K 8644, PD98059, staurosporine, wortmannin, DRB, or gemcitabine, which depressed cell proliferation [179]. Extracellular regucalcin did not induce apoptotic cell death in MiaPaCa-2 cells in vitro [179]. Thus, extracellular regucalcin may have suppressive effects on the proliferation of human pancreatic MiaPaCa-2 cells mediated through various signaling pathways in vitro.

The proliferation of bone metastatic human breast cancer MDA-MB-231 cells has been shown to be suppressed by culturing with extracellular regucalcin [181]. In this study, extracellular regucalcin did not induce apoptotic cell death of MDA-MB-231 cells in vitro [181].

Interestingly, extracellular regucalcin has been found to have suppressive effects on the growth, colony formation, migration, invasion, and adhesion of metastatic human prostate cancer PC-3 and DU-145 cells in vitro, as reported in [182]. The suppressive effects of extracellular regucalcin may result from a decrease in levels of multiple signaling proteins, such as Ras, phosphatidylinositol-3 kinase, MAP kinase, mTOR, RSK-2, caveolin-1, and integrin β1, in PC-3 cells [182]. This study demonstrated in vitro inhibition of metastatic activity by extracellular regucalcin.

Extracellular regucalcin suppressed proliferation in vitro independent of SK-OV-3 cell death [177]. The proliferation of SK-OV-3 cells was enhanced by culturing with EGF [177], which was then suppressed by extracellular regucalcin [177]. This suggests that the binding of EGF to its receptors in the plasma membranes of cells is antagonized by extracellular regucalcin. However, extracellular regucalcin did not reduce the levels of EGF receptor protein [177]. Extracellular regucalcin has potential for suppressing cell proliferation through various signaling pathways, particularly those related to EGF signaling proteins. This is accomplished through the targeting of specific proteins in a mechanistic manner.

Furthermore, extracellular regucalcin exhibited inhibitory effects on the growth of human osteosarcoma Saos-2 cells in vitro [161]. Incubation with extracellular regucalcin at 1 and 10 nM led to decreased colony formation and proliferation of Saos-2 cells [161]. Importantly, extracellular regucalcin did not induce cell death in Saos-2 cells [161]. Extracellular regucalcin was observed to decrease the levels of several molecules including Ras, PI3K, Akt, MAP kinase, phosphor-MAP kinase, STAT3, NF-κB p65, and β-catenin, while increasing the levels of p21, which is known to suppress cell proliferation. This suggests that extracellular regucalcin may have potential therapeutic benefits in treating diseases that involve exacerbated cell proliferation.

As previously noted, extracellular regucalcin has been shown to inhibit the growth of different human cancer cell types in vitro without affecting cell death. It is highly likely that the extracellular regucalcin produced within the tissues plays a vital role in inhibiting cancer cell growth. Therefore, extracellular regucalcin may potentially function as a suppressive factor similar to cytokines in suppressing cell growth.

The mechanism underlying extracellular regucalcin’s suppression of cancer cell growth involves blocking different intracellular signaling pathways that participate in cell proliferation. This is illustrated in Figure 4. Extracellular regucalcin binds to the plasma membranes of cancer cells. The bound regucalcin can cause signal transduction, generating a factor that suppresses intracellular signaling pathways associated with transcription in the nucleus of cancer cells. Additionally, the binding of regucalcin to plasma membranes may potentially trigger cellular internalization and hence, influence cell signaling processes, resulting in the inhibition of cell proliferation.

Extracellular regucalcin, which is increased in the cancer microenvironment, potentially suppresses carcinogenesis in various tissues. Additionally, extracellular regucalcin may prevent adhesion, invasion, and migration of cancer cells, thereby blocking their metastasis.

## 6. Conclusions and Perspectives

Regucalcin was originally discovered in 1978 as a novel calcium-binding protein lacking the EF-hand motif of the calcium-binding domain [7,8]. Regucalcin was shown to play a critical role in maintaining intracellular calcium homeostasis and as an inhibitory protein of calcium signaling, which plays a pivotal role in the regulation of cell functions [5,6,8]. Subsequently, this protein was found to play a multifunctional role in maintaining cell homeostasis in various cell types [5]. In addition, regucalcin has been shown to play a pathophysiological role in various diseases [5,6,25,26,27], including human cancer [34,35,36,37,38,39,40].

As presented in this literature review, regucalcin—a protein that plays an indispensable role in controlling cell growth—has been identified as a potential suppressor in the development of human carcinogenesis. The expression of regucalcin is notably diminished in diverse tissues of cancer patients, such as the liver [32,34,63,64,65], pancreas [35], colon [38], lung [37,82,83,84], kidney [39], breast [36], prostate [40,96], cervical [149,150], and melanoma [154], as investigated by our group and other researchers. A greater expression of regucalcin in tumor tissues extends the survival of patients with different cancer types. Recent studies have shown that extracellular regucalcin, found in the extracellular fluids of various tissues, suppresses the growth of several types of human cancer cells, such as HepG2 liver cancer cells [180], MiaPaCa-2 pancreatic cancer cells [179], MDA-MB-231 breast cancer cells [181], PC-3 prostate cancer cells [182], SK-OV-3 ovarian cancer cells [177], and Saos-2 osteosarcoma cells [161]. Extracellular regucalcin contributes to suppressing cancer cell growth in the cancer microenvironment. These studies suggest that regucalcin has potential as a suppressor in the development of human malignancies.

A recent study demonstrated that inflammatory macrophages inhibit the growth of human prostate cancer cells that have spread to the bones through intracellular signaling of TNF-α and IL-6 produced by macrophages [17]. The action of TNF-α and IL-6 is mediated through transcription factors, specifically NF-κB or STAT3 [17]. Moreover, these transcription factors have been observed to increase the activity of regucalcin gene expression [17]. Inflammatory macrophages may trigger prostate cancer cell loss through NF-κB, STAT3, or regucalcin-linked processes (17). Figure 1 shows that regucalcin gene expression is enhanced by various transcription factors, including HIF-1α and β-catenin. Intracellular regucalcin mediates the cellular signaling effects of various signaling factors that regulate cell function. Further research is necessary to explore intracellular regucalcin’s role as a mediator of cellular signaling.

The role of regucalcin in the prevention and management of human cancer is demonstrated in Figure 5. Reduced expression of regucalcin in tumor tissues contributes to the development of carcinogenesis in multiple tissues and affects patient prognosis. Regucalcin gene expression and protein levels can be elevated by several factors, including hormones, intracellular signaling factors, and transcription factors. Furthermore, various pathophysiological conditions, aging, and environmental components may repress the expression of the regucalcin gene. Although epigenetic modification has been proposed to contribute to the decreased expression of regucalcin, the mechanism underlying its downregulation in tumor tissues requires elucidation through functional studies [82].

In conclusion, the findings demonstrate that regucalcin could be a valuable biomarker for diagnosing and treating various types of human cancer. These results indicate the potential clinical significance of regucalcin as a novel diagnostic and therapeutic target in cancer research. Overexpression of regucalcin in tumor cells has been shown to inhibit cancer cell growth, while its underexpression is associated with early stages of cancer progression. Since regucalcin expression is decreased in tumor tissues, increasing its expression may be crucial. This could help regulate cancer development through the elevated expression of regucalcin. Figure 5 indicates that therapeutic benefits can result from higher regucalcin levels in tumor tissues and cells, which can be achieved by enhancing the expression of the regucalcin gene through hormonal, nutritional, and chemical therapies. Objectively, delivery of regucalcin genes into tumor tissues may exhibit a depressive impact on tumorigenesis. Moreover, administering exogenous regucalcin treatment may play a role in tumor development suppression within the cancer microenvironment. However, clinical application of this therapy may present various challenges. Further research is required to establish the clinical potential of regucalcin as a suppressor of human cancer.

## Figures and Tables

**Figure 1 cancers-15-05489-f001:**
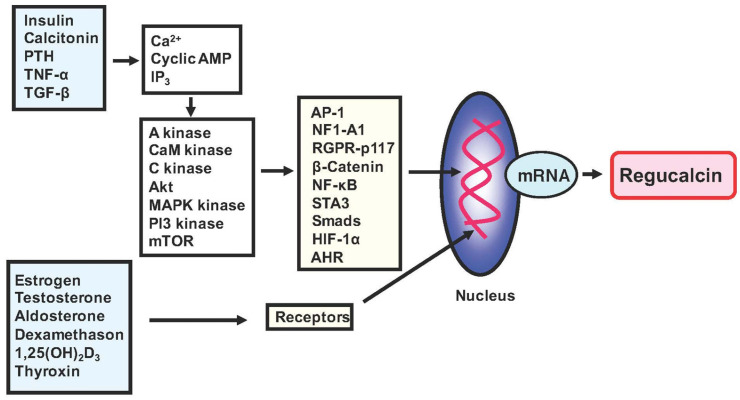
The expression of regucalcin is regulated by various signaling factors implicated in the action of peptide hormones, steroid hormones, and other factors. The transcription factors AP-1, NF1-A1, RGPR-p117, and others are involved in the expression of the regucalcin gene. These transcription factors are transported from the cytoplasm to the nucleus through a mechanism mediated via intracellular signaling factors. These signaling factors include cyclic AMP-dependent protein kinase (A kinase), Ca^2+^-calmodulin-dependent protein kinase (CaM kinase), and protein kinase C (C kinase). These factors are coupled to the signaling processes of various factors. Steroid hormones directly bind to receptors in the cytoplasm and nucleus, increasing the expression of the regucalcin gene through transcription factors. Additionally, the promoter activity of the regucalcin gene is enhanced in the nucleus.

**Figure 2 cancers-15-05489-f002:**
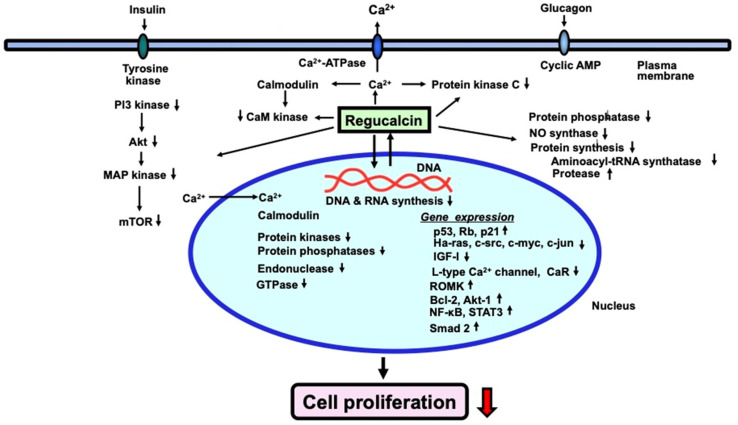
Regucalcin inhibits proliferation in various types of normal and cancer cells. Within cells, intracellular regucalcin reduces the activity of several enzymes involved in signaling, such as Ca^2+^/calmodulin-dependent enzymes, protein kinases, and protein phosphatases in the cytoplasm. Additionally, regucalcin activates cysteinyl protease and aminoacyl-tRNA synthetase, leading to a decrease in protein production. Cytoplasmic regucalcin is translocated to the nucleus through protein kinase-related signaling. Nuclear regucalcin inhibits both Ca^2+^-dependent and -independent protein kinase and protein phosphatase, and suppresses nuclear DNA and RNA synthesis. Overexpressed regucalcin obstructs the G1 and G2/M phases of the cell cycle. Therefore, regucalcin’s suppression of cell proliferation occurs by regulating various signaling processes. The black upward arrow signifies upregulation, while the black and red downward arrows denote downregulation.

**Figure 3 cancers-15-05489-f003:**
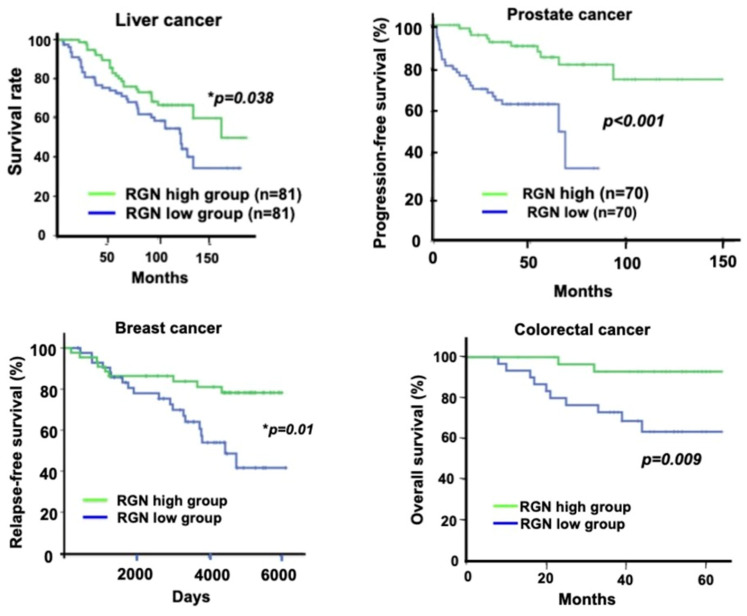
Regucalcin may have a role in suppressing human cancer. Low expression levels of the regucalcin gene and protein seem to be associated with unfavorable outcomes in patients with various types of human cancer. The patients were divided into two groups based on their regucalcin gene expression, high and low. The high regucalcin gene expression group exhibited a statistically significant difference (*p*-value) compared to the low group. The Kaplan-Meier curve showed a significant increase in the survival rate of cancer patients in the high regucalcin gene expression group compared to the low expression group. Abbreviations: RGN, regucalcin.

**Figure 4 cancers-15-05489-f004:**
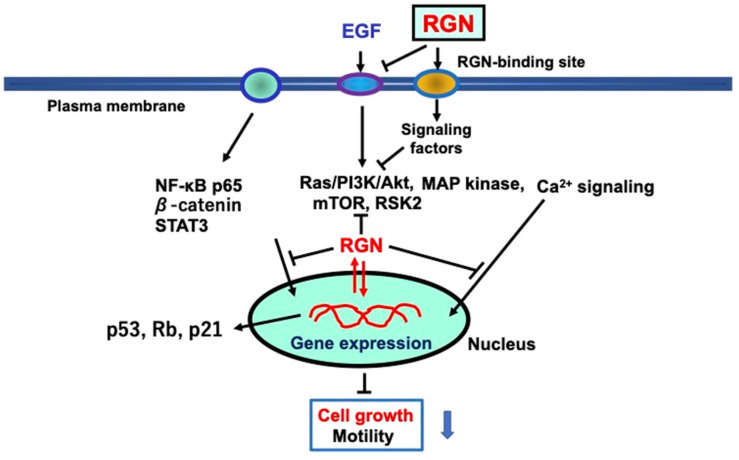
The underlying mechanism by which extracellular regucalcin suppresses the proliferation of human cancer cells in vitro. Extracellular regucalcin blocks various EGF-related signaling pathways by targeting EGF receptors in cells. In addition, extracellular regucalcin may bind to putative regucalcin-binding sites on plasma membranes to transmit signals to cells. In addition, intracellular regucalcin may suppress other signaling processes and the levels of transcription factors. In particular, extracellular regucalcin increases the levels of p53, Rb, and p21, which are suppressors of tumorigenesis, suggesting an effect on nuclear function. Extracellular regucalcin may affect various molecules in cancer cells, leading to the promotion of tumorigenesis. Extracellular regucalcin may play a critical role as a suppressor in the cancer microenvironment. Abbreviations: RGN; regucalcin; EGF; epidermal growth factor. The black down arrow indicates downregulation. The blue down arrow indicates downregulation.

**Figure 5 cancers-15-05489-f005:**
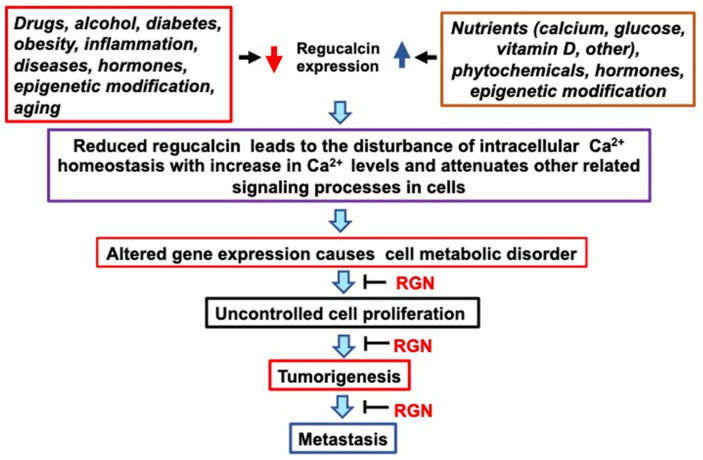
The role of regucalcin in the development and management of human cancer is examined. The expression of the regucalcin gene is influenced by different factors, such as hormonal, nutritional, and chemical factors. Altered regucalcin gene expression can cause various metabolic disorders, leading to uncontrolled cell proliferation, tumorigenesis, and metastasis. This process can be blocked by increasing the levels of regucalcin. Preventive and therapeutic efficacy of cancer can be brought by increasing regucalcin levels in tissues and cells through epigenetic modification, nutritional factors, and natural chemical treatment. Abbreviations: RGN; regucalcin. The red down arrow indicates downregulation. The blue up arrow indicates upregulation.

## Data Availability

The datasets used during the present study are available from the corresponding author upon reasonable request.

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
