# Peer review of "Regucalcin Is a Potential Regulator in Human Cancer: Aiming to Expand into Cancer Therapy"

_cancers, 2023, doi:10.3390/cancers15225489_

Round 1
Reviewer 1 Report
Comments and Suggestions for Authors
The manuscript presented by Masayoshi Yamaguchi is scientifically interesting, and worth publication concerning the focus and relevance of the regucalcin role in cell function. It reunites important information describing the putative tumor suppressor role of regucalcin in several cancer types, and the recent findings on the subject, proposing this protein as a treatment target. However, the assumption of “treatment with the regucalcin gene delivery and its protein injection to patients” is highly speculative and needs to be adequately discussed on the possibilities and difficulties that could be associated with it.
Moreover, the manuscript needs extensive writing revision for increased accuracy and clarity.
Also, figure legends need revision. Figure 5 does not show a therapeutic strategy.
Sub-headings in section 4, can only be cancer types 4.1. Liver cancer, 4.2. lung cancer, etc…
Comments on the Quality of English LanguageThe manuscript needs extensive writing revision for increased accuracy and clarity.
Author Response
Comments of Reviewer
Reviewer 1:
Comments and Suggestions for Authors;
The manuscript presented by Masayoshi Yamaguchi is scientifically interesting, and worth publication concerning the focus and relevance of the regucalcin role in cell function. It reunites important information describing the putative tumor suppressor role of regucalcin in several cancer types, and the recent findings on the subject, proposing this protein as a treatment target.
- However, the assumption of “treatment with the regucalcin gene delivery and its protein injection to patients” is highly speculative and needs to be adequately discussed on the possibilities and difficulties that could be associated with it.
Reply of the author:
Thank you very much for your kind suggestion. As indicated by reviewer, the last paragraph of “Section 5” was revised. This author discussed on the possibilities and difficulties of clinical application of regucacin.
- Moreover, the manuscript needs extensive writing revision for increased accuracy and clarity.
Reply of the author:
Thank you very much for your kind suggestion.
This author extensively revised the manuscript by using editing soft,
(https://www.deepl.com/write) and Grammarly.
Moreover, the manuscript was checked by a native English scientist.
- Also, figure legends need revision. Figure 5 does not show a therapeutic strategy.
Reply of the author:
Thank you very much for your kind suggestion. The author revised figure legends. Especially, the legend title of Figure 5 was revised as indicated by the reviewer as “Involvement of regucalcin in the causation and control of human cancer”.
- Sub-headings in section 4, can only be cancer types 4.1. Liver cancer, 4.2. lung cancer, etc…
Reply of the author:
Thank you very much for your kind suggestion. The author revised all sub-headings in section 4, as indicated by the reviewer.
- Comments on the Quality of English Language. The manuscript needs extensive writing revision for increased accuracy and clarity.
Reply of the author:
Thank you very much for your kind suggestion.
As described in number 5, the author extensively revised the manuscript by using editing soft,
(https://www.deepl.com/write) and Grammarly.
Moreover, the manuscript was checked by a native English scientist.
Reviewer 2 Report
Comments and Suggestions for Authors
Please find an attachment PDF file (cancers 2023 Comment to Author.pdf).
The reviewer noted their comments there.

There are a few minor corrections.
Author Response
Reviewer 2:
Comments to Author;
Regarding the function of regucalcin, recently, the inhibitory function on the growth of cancer cells, has been attracting attention. The author and his colleagues have previously demonstrated that regucalcin is effective in suppressing the proliferation of various human cells/tissues, such as colorectal carcinoma, liver HepG2, lung adenocarcinoma A549, breast cancer MDA-MB-231, renal cell carcinoma, pancreatic cancer MIA PaCa2, and prostate cancer PC-3 and DU-145. This reviewer understands that the present manuscript is one of the culminations of regucalcin research in cancer by the author. This review manuscript is sure to do a lot of contribution to the researcher in this field. Therefore, this reviewer strongly recommends that the manuscript be published in cancers. However, need to correct minor points before the paper can be accepted for publication.
The reply of author:
This author really appreciates the comments of this reviewer. As indicated by reviewer, all items with type miss were corrected.
- Line 41. cell communication [1].: Add a period at the end of the sentence.
Reply of author: This was corrected.
- Line 44. pleiotrophy → pleiotropy: It is more general.
Reply of author: This was corrected.
- Line 90. kidney [58]: All are disease names except kidney.
Reply of author: This was corrected “kidney failure”.
- Figure 1. Two “insulin” are indicated in the upper left frame. Delete one or the other.
Reply of author: This was corrected in Figure 1.
- Line 107. various tra itnscription factors → various transcription factors
Reply of author: This was corrected in Fgure 1 legends.
- 109-110. Abbreviations: RGN; regucalcin. → RGN is not indicated in the figure and should be deleted.
Reply of author: This was corrected.
- Line 262. GEO: Define this abbreviation.
Reply of author: This was corrected.
- Line 142. cells .: less the space between “cell” and period. Figure 3. Resolution is insufficient.
Reply of author:
This author used 300dpi of the resolution in Figure 3. This is used usually, and I liked this resolution. Figure 3 included many figures. The author revised Figure 3 with large scale. Therefore, the author deleted the data of lung cancer. This was white and black figure compared to other cancer data. I used 6 cancer data in the revised figure 3. Thank you very much for your kind consideration.
- Line 360. Regucalcin: Font size is large.
Reply of author: This was corrected.
- Line 367. Regucalcin was shown to: Font size is large.
Reply of author: This was corrected.
- Line 373. [133]. [134]. → [133, 134].
Reply of author: This was corrected.
- Line 398. leading to control of cell growth [71].: Delete this sentence.
Reply of author: This sentence was corrected to “transcription 3 (STAT3), leading to control of cell growth [71].”
- Line 540. [203, 204].: Font size is large.
Reply of author: This was corrected.
- Line 690. types.. → types.
Reply of author: This was corrected.
- There are two No.84. Check the reference number
Reply of author: This was corrected.
Reviewer 3 Report
Comments and Suggestions for Authors
This paper is well written. This paper would be attracted interesting for the reader of the Cancers.
Author Response
Comments and Suggestions for Authors
This paper is well written. This paper would be attracted interesting for the reader of the Cancers.
The reply of author:
Thank you very much for your kind comments. The author did not need the revision for the comments of reviewer.
Round 2
Reviewer 1 Report
Comments and Suggestions for Authors
The authors have answered the reviewer's comments and improved the manuscript.
Comments on the Quality of English LanguageThe authors have answered the reviewer's comments concerning language issues and improved the manuscript.
Reviewer 2 Report
Comments and Suggestions for Authors
The comments noted by this reviewer habve been adequately addressed .